# Sociodemographic Disparities in Adults with Kidney Failure: A Meta-Analysis

**DOI:** 10.3390/diseases12010023

**Published:** 2024-01-12

**Authors:** Ioannis Bellos, Smaragdi Marinaki, Evangelia Samoli, Ioannis N. Boletis, Vassiliki Benetou

**Affiliations:** 1Department of Hygiene, Epidemiology and Medical Statistics, Medical School, National and Kapodistrian University of Athens, 11527 Athens, Greece; esamoli@med.uoa.gr (E.S.); vbenetou@med.uoa.gr (V.B.); 2Department of Nephrology and Renal Transplantation, Laiko General Hospital, National and Kapodistrian University of Athens, 11527 Athens, Greece; smaragdimarinaki@yahoo.com (S.M.); inboletis@gmail.com (I.N.B.)

**Keywords:** dialysis, transplantation, equity, race, sex

## Abstract

This meta-analysis aims to assess current evidence regarding sociodemographic disparities among adults with kidney failure. Medline, Scopus, Web of Science, CENTRAL, and Google Scholar were systematically searched from inception to 20 February 2022. Overall, 165 cohort studies were included. Compared to White patients, dialysis survival was significantly better among Black (hazard ratio—HR: 0.68; 95% CI: 0.61–0.75), Asian (HR: 0.67; 95% CI: 0.61–0.72) and Hispanic patients (HR: 0.80; 95% CI: 0.73–0.88). Black individuals were associated with lower rates of successful arteriovenous fistula use, peritoneal dialysis and kidney transplantation, as well as with worse graft survival. Overall survival was significantly better in females after kidney transplantation compared to males (HR: 0.87; 95% CI: 0.84–0.90). Female sex was linked to higher rates of central venous catheter use and a lower probability of kidney transplantation. Indices of low SES were associated with higher mortality risk (HR: 1.22, 95% CI: 1.14–1.31), reduced rates of dialysis with an arteriovenous fistula, peritoneal dialysis and kidney transplantation, as well as higher graft failure risk. In conclusion, Black, Asian and Hispanic patients present better survival in dialysis, while Black, female and socially deprived patients demonstrate lower rates of successful arteriovenous fistula use and limited access to kidney transplantation. PROSPERO registration: CRD42022300839.

## 1. Introduction

Kidney failure represents a rising global public health concern, as it exerts a significant impact on patients’ quality of life and is associated with high mortality rates [1]. More than 80% of patients receiving renal replacement therapy come from high-income countries, due to their aging population and satisfactory access to healthcare services. On the contrary, access to dialysis and kidney transplantation remains limited in low- and middle-income countries and a substantial proportion of patients with kidney failure die even without supportive care [2].

Health disparities are health differences that adversely affect socially disadvantaged populations. They are defined as systematic, potentially avoidable health differences according to, among others, race/ethnicity, gender, geographic differences and socioeconomic status [3]. The risks of chronic kidney disease are not equally distributed in the population since its incidence and complications present remarkable variation across ethnic, racial and socioeconomic groups [4].

Specifically, the risk of reaching kidney failure has been estimated to be significantly higher among Black and Hispanic patients by 3.4 and 1.3 times, respectively [5]. In addition, Asian Americans and Pacific Islanders have been shown to present a higher risk of early stage kidney disease, while native Hawaiian individuals have been proposed to be at greater risk of severe kidney disease forms [6]. Racial discrimination may be associated with kidney disease progression through its potential effects on social determinants of health, such as access to education and healthcare, while it may exert biological impact by contributing to increased allostatic load and epigenetic changes that may be implicated in altered hormone metabolism, albuminuria and kidney function decline [7,8]. Notably, Black individuals have exhibited a higher risk of albuminuria and rapid kidney function decline that may be partially explained by social and environmental components leading to increased rates of comorbidities, whereas genetic variations in the APOL1 gene have also been associated with younger age at hemodialysis initiation [9,10]. A low socioeconomic status has been also suggested to increase the risk of kidney disease since it has been associated with a lower estimated glomerular filtration rate, greater albuminuria and higher rates of progressive kidney function loss [11].

Growing research interest exists about kidney health inequities; although, evidence regarding disparities in patients reaching end-stage kidney disease remains scattered. The present study aims to accumulate current literature evidence and assess the potential association of sociodemographic parameters with the management and outcomes of patients with kidney failure. In particular, a systematic review and meta-analysis was conducted, investigating whether race/ethnicity, sex and socioeconomic status are associated with the choice of renal replacement therapy modality, the probability of kidney transplantation and the graft survival, as well as the survival of dialysis patients and transplant recipients.

## 2. Materials and Methods

### 2.1. Study Design

The meta-analysis was designed following the PRISMA (Preferred Reporting Items for Systematic Reviews and Meta-Analyses) guidelines [12]. The protocol was prospectively registered in PROSPERO (CRD42022300839). Ethical approval was not required since no patient data were collected.

### 2.2. Eligibility Criteria

The population of the study consisted of adult patients (age ≥ 18 years) with kidney failure (estimated glomerular filtration rate < 15 mL/min/1.73 m^2^), receiving renal replacement therapy in the form of hemodialysis, peritoneal dialysis or kidney transplantation. The main exposures were race/ethnicity (White, Black, Asian, Indigenous and Hispanic), biological sex and socioeconomic status. Socioeconomic status was defined based on income, education and/or occupation status. White patients, males and those with a high socioeconomic status served as the refer-ence groups. The outcomes of interest were dialysis-related and kidney transplantation-related ones. Specifically, the dialysis-related outcomes concerned the dialysis modality, the choice of vascular access for hemodialysis and patient survival, including the following:-Start of hemodialysis with an arteriovenous fistula (AVF) or graft (AVG) vs. a central venous catheter (CVC);-Start of hemodialysis with an AVF vs. an AVG;-Successful AVF use;-Primary patency of AVF;-Transition from CVC to AVF or AVG;-Use of home dialysis;-Use of peritoneal dialysis;-All-cause mortality.

The transplantation-related outcomes evaluated the probability of receiving a kidney allograft and the type of transplantation, as well as the graft and patient survival, including the following:-Waitlisting;-Receipt of any kidney allograft;-Deceased-donor transplantation;-Living-donor transplantation;-Preemptive transplantation;-Graft survival;-All-cause mortality.

Prospective and retrospective cohort studies were deemed eligible. Case–control, descriptive and cross-sectional studies were excluded.

### 2.3. Literature Search

Medline, Scopus, Web of Science and Cochrane Central Register of Controlled Trials (CENTRAL) were systematically searched without applying date or language restrictions. Google scholar and the reference lists of the included studies (“snowball” method [13]) were additionally searched to identify potential missing articles that were not recognized by the primary search. The date of the last search was set at 20 February 2022. The search algorithm was based on the combination of keywords and MeSH (Medical Subject Headings) terms and is schematically presented in Appendix A.

### 2.4. Study Selection

The process of study selection was performed following three stages. At first, the titles and abstracts of all electronic articles identified through the literature search were screened. Subsequently, the articles that were presumed to be potentially eligible were retrieved in full-text form. Then, the studies that were assessed to meet the eligibility criteria of the meta-analysis were included. Studies that did not report the pre-defined exposures and outcomes or met any of the exclusion criteria were not included. The selection of studies was conducted by two researchers (IB, VB) independently, resolving any discrepancies through the consensus of authors.

### 2.5. Data Collection

Data extraction was performed by filling pre-defined forms including the following information: reference details, country, study design, population, data source, time period, sample size, patients’ sex and percentage of diabetes mellitus. In addition, the data regarding the meta-analysis exposures and outcomes were collected. In case of missing data, the corresponding authors of the original articles were contacted, requiring the necessary information. The process of data collection was conducted by two researchers (IB, SM) and possible disagreements were resolved by consensus.

### 2.6. Risk of Bias Assessment

The risk of bias was judged by applying the Risk Of Bias In Non-randomized Studies of Interventions (ROBINS-I) tool for each included study. The tool was adjusted for exposure studies. In particular, the risk of bias was assessed as low, moderate, high or critical in the domains of confounding, participants selection, exposure classification, deviations from intended exposures, missing data, outcome measurement and selection of the reported result [14]. Quality assessment was performed by two researchers independently (IB, ES), resolving any potential discrepancies by consensus.

### 2.7. Data Analysis

Statistical analysis was conducted in R-4.0.5 (package: “metafor” [15]). Statistical significance was defined at the level of 5%. Odds ratios (OR) or hazard ratios (HR) along with their 95% confidence intervals (CI) were extracted from the multivariate models (where available) of original studies; in case multivariate models were not available or estimates were not directly reported, the crude OR or HR were extracted or calculated. White individuals, male sex and high socioeconomic status constituted the reference groups. In particular, concerning socioeconomic status, the lowest category was compared to the highest one. Meta-analysis was performed in the case of 2 or more studies per outcome. Due to the expected inter-study heterogeneity, random-effects models were fitted using the restricted maximum likelihood method. Statistical heterogeneity was quantified using the inconsistency index (I^2^) [16], with values >50% indicating the existence of remarkable heterogeneity. The 95% prediction intervals were calculated to provide estimates of the effects to be expected by future studies in the field [17]. Subgroup analysis was performed based on study location, risk of bias, type of dialysis population (hemodialysis or peritoneal dialysis patients) and the definition of socioeconomic status. *p*-values for subgroup differences were generated by conducting meta-regression analysis. Sensitivity analysis was conducted to assess exclusively studies that took place in the USA. Funnel plots were constructed and visually assessed. The significance of Egger’s regression test (*p*-value < 0.10) was taken into account in the case of at least 10 included studies [18].

### 2.8. Certainty of Evidence

The GRADE (Grading of Recommendations, Assessment, Development and Evaluations) approach was implemented to assess the certainty of the existing evidence. Specifically, the certainty was appraised as high, moderate, low or very low by taking into consideration the following domains: study limitations, indirectness, imprecision, inconsistency and publication bias [19].

## 3. Results

### 3.1. Study Selection

The process of study selection is schematically illustrated in Figure 1. After the removal of duplicate electronic papers, 4900 articles were screened for eligibility and 284 of them were retrieved as full texts. Then, 119 studies were excluded due to the lack of reporting of any of the outcomes of interest (n = 46), the inclusion of overlapping populations with other included studies (n = 27), the assessment of different exposures (n = 22), the inclusion of pediatric participants (n = 18), the evaluation of immigration (n = 3) or due to their descriptive study design (n = 3) (Appendix A). As a result, a total of 165 studies were included in this review.

### 3.2. Included Studies

The baseline characteristics of the included studies are described in Appendix A. Table 1 summarizes their main methodological features. The main country of study conduct was the USA (70 studies), followed by Canada (14 studies), the United Kingdom (13 studies) and Australia (11 studies). Forty-six studies were prospective cohorts, while the remaining implemented a retrospective cohort design. The United States Renal Data System (USRDS) was the main data registry, used by 31 studies. The definitions of income, education and occupation used in the included studies are described in Appendix A. The risk of bias assessment of the included studies is presented in Appendix A. Overall, the risk of bias was evaluated to be low in 105 studies, moderate in 53 and serious in 8 studies. No study was judged to be at critical risk. Specifically, the risk of confounding was moderate in 36 and serious in 8 studies, due to the lack of adjustment for important covariates. A moderate risk of selection bias was recognized in 14 studies, while a moderate risk of bias in the domain of classification of exposures was detected in 12 studies due to self-classification of the exposure (race or socioeconomic status) by the participants. No information regarding the exact risk of missing data was available for 52 studies due to the incomplete reporting of reasons for participant exclusion or the lack of data regarding the exposure and outcome of patients excluded from the study during follow-up.

### 3.3. Dialysis

The meta-analysis results along with the certainty of evidence concerning the association of race/ethnicity, sex and socioeconomic status with dialysis-related outcomes are depicted in Figure 2. Forest plots are illustrated in Appendix A and funnel plots in Appendix A. The outcomes of subgroup analyses are presented in Appendix A, while the results of sensitivity analyses are exhibited in Appendix A. A summary of evidence is provided in Appendix A.

#### 3.3.1. Black Individuals

Black individuals were associated with a significantly higher probability of starting hemodialysis with an AVF or AVG than a CVC (OR: 1.08; 95% CI: 1.07–1.10; two studies; high certainty). However, Black individuals presented a significantly lower probability of hemodialysis via AVF compared to AVG (OR: 0.58; 95% CI: 0.51–0.66; four studies; high certainty), as well as lower rates of successful AVF use (OR: 0.89; 95% CI: 0.84–0.94; two studies; high certainty) and worse primary AVF patency (HR: 0.87; 95% CI: 0.86–0.88; two studies; high certainty). The probability of transition from CVC to AVF/AVG was similar between Black and White individuals (OR: 1.06; 95% CI: 0.92–1.22; two studies; low certainty). Compared to White people, Black individuals had similar rates of initiating home dialysis (OR: 1.00; 95% CI: 0.83–1.21; two studies; high certainty), but a significantly lower probability of peritoneal dialysis use (OR: 0.65; 95% CI: 0.49–0.88; five studies; low certainty). Mortality rates were significantly lower among Black than White patients (HR: 0.68; 95% CI: 0.61–0.75; 18 studies; moderate certainty). This association was evident in both hemodialysis and peritoneal dialysis patients and remained significant in North America, South America and Europe.

#### 3.3.2. Asian Individuals

The probability of initiating hemodialysis with an AVF/AVG rather than a CVC was similar between Asian and White individuals (OR: 1.04; 95% CI: 0.76–1.4; two studies; moderate certainty). Home dialysis use did not differ between them also (OR: 0.86; 95% CI: 0.52–1.42; two studies; low certainty); although, Asians were more likely to implement peritoneal dialysis (OR: 1.28; 95% CI: 1.00–1.64; three studies; low certainty), especially in Canada. Asians were associated with significantly better overall survival rates during dialysis (HR: 0.67; 95% CI: 0.61–0.72; 14 studies; moderate certainty). This survival advantage applied to both hemodialysis and peritoneal dialysis patients and was not affected by the location or the risk of bias of the studies. Importantly, Asian patients also presented better dialysis survival in one study conducted exclusively in Asia (China).

#### 3.3.3. Indigenous People

The rates of using an AVF/AVG for hemodialysis did not differ between Indigenous and White patients (OR: 0.93; 95% CI: 0.82–1.07; three studies; moderate certainty). Nonetheless, subgroup analysis indicated that Indigenous patients in Oceania (Australian Aboriginal, Torres Strait Islander and Māori peoples) presented a significantly lower probability of hemodialysis via an AVF or AVG (OR: 0.85; 95% CI: 0.82–0.89). No significant difference was estimated between the use of AVF and AVG (OR: 0.93; 95% CI: 0.42–2.09; two studies; low certainty). Overall, peritoneal dialysis implementation rates were similar between Indigenous and White populations (OR: 0.83; 95% CI: 0.61–1.14; two studies; low certainty), although the study conducted in Canada demonstrated a significantly lower probability of peritoneal dialysis for Aboriginal patients. Mortality rates were similar between Indigenous and White populations (HR: 1.02; 95% CI: 0.85–1.23; 11 studies; low certainty), an observation present both in North America and Oceania.

#### 3.3.4. Hispanic Ethnicity

Hispanic ethnicity was associated with significantly better survival rates among dialysis patients (HR: 0.80; 95% CI: 0.73–0.88; eight studies; moderate certainty). This outcome was stable in the subgroup analysis, remaining statistically significant in studies both in North and South America, as well as among hemodialysis and peritoneal dialysis patients.

#### 3.3.5. Female Sex

The probability of hemodialysis via an AVF or AVG was significantly lower among female patients (OR: 0.81; 95% CI: 0.69–0.95; five studies; low certainty). Among those with permanent access, females were less likely to have an AVF than an AVG (OR: 0.93; 95% CI: 0.42–2.09; five studies; high certainty). The difference between males and females was not significant regarding successful AVF use (OR: 0.60; 95% CI: 0.40–1.02; three studies; very low certainty); although, the female sex was linked to significantly worse primary AVF patency (HR: 0.73; 95% CI: 0.53–0.96; four studies; low certainty). The probability of transition from CVC to AVF/AVG was also lower among female patients (OR: 0.92; 95% CI: 0.85–1.00; two studies; moderate certainty). The overall peritoneal dialysis rates were similar between the two sexes (OR: 1.07; 95% CI: 0.92–1.25), though presenting significant variation across countries. Mortality rates did not differ between males and females (HR: 0.99; 95% CI: 0.95–1.04; 26 studies; low certainty). This finding was consistent across continents, as well as in both hemodialysis and peritoneal dialysis populations.

#### 3.3.6. Low Socioeconomic Status

Compared to patients with high socioeconomic status, those with a lower one presented significantly lower rates of hemodialysis using an AVF or AVG (OR: 0.81; 95% CI: 0.65–1.00; five studies; low certainty). Low socioeconomic status was also associated with significantly worse primary AVF patency (HR: 0.43; 95% CI: 0.39–0.47; three studies; moderate certainty), lower rates of transition from CVC to AVF/AVG (OR: 0.80; 95% CI: 0.73–0.88; two studies; moderate certainty) and use of peritoneal dialysis (OR: 0.56; 95% CI: 0.44–0.70; five studies; low certainty). Dialysis patients with low socioeconomic status were estimated to be at significantly higher risk of mortality (HR: 1.22; 95% CI: 1.14–1.31; 23 studies; very low certainty). This outcome remained stable across different dialysis modalities and socioeconomic status definitions; although, it was characterized by inter-study heterogeneity and was deemed as prone to publication bias.

### 3.4. Kidney Transplantation

The results of the meta-analysis and the GRADE assessment regarding the kidney transplantation-related endpoints are illustrated in Figure 3. Forest plots are provided in Appendix A and funnel plots in Appendix A. The subgroup analyses are presented in Appendix A and the outcomes of sensitivity analyses in Appendix A. A summary of evidence is presented in Appendix A.

#### 3.4.1. Black Individuals

Black patients were associated with a significantly lower probability of waitlisting for kidney transplantation (OR: 0.80; 95% CI: 0.70–0.91; seven studies; moderate certainty), as well as with significantly lower rates of kidney transplantation (OR: 0.70; 95% CI: 0.49–0.99; five studies; low certainty) compared to White patients. Specifically, Black patients presented lower rates of living-donor transplantation (OR: 0.40; 95% CI: 0.31–0.50; seven studies; high certainty) but comparable rates of deceased-donor transplantation with White patients (OR: 0.77; 95% CI: 0.51–1.16; six studies; low certainty). Preemptive transplantation was significantly less frequent among Black patients (OR: 0.46; 95% CI: 0.44–0.49; two studies; high certainty). Graft survival was estimated to be significantly worse among Black patients (HR: 1.44; 95% CI: 1.27–1.63; eight studies; moderate certainty). Mortality rates were similar among Black and White transplant recipients (HR: 0.96; 95% CI: 0.83–1.12; four studies; moderate certainty).

#### 3.4.2. Asian Individuals

The probability of waitlisting was estimated to be higher among Asians than White people (OR: 1.25; 95% CI: 1.08–1.44; two studies; moderate certainty). However, the overall probability of kidney transplantation was significantly lower in Asians (OR: 0.59; 95% CI: 0.42–0.82; two studies; low certainty). Specifically, Asian individuals were associated with significantly lower rates of living-donor transplantation (OR: 0.42; 95% CI: 0.28–0.62; four studies; low certainty) but similar rates of deceased-donor transplantation compared to White individuals (OR: 0.89; 95% CI: 0.65–1.22; three studies; low certainty).

#### 3.4.3. Indigenous People

Indigenous patients presented a significantly lower probability of waitlisting (OR: 0.63; 95% CI: 0.44–0.91; three studies; low certainty), both in the USA and Australia. The overall transplantation rates were significantly lower among Indigenous populations (OR: 0.37; 95% CI: 0.27–0.50; seven studies; moderate certainty), in both the USA and Australia. Indeed, Indigenous patients showed a significantly lower probability of both living-donor (OR: 0.31; 95% CI: 0.22–0.43; seven studies; low certainty) and deceased-donor transplantation (OR: 0.39; 95% CI: 0.29–0.53; six studies; low certainty). Graft survival did not differ significantly between Indigenous and White patients (HR: 1.42; 95% CI: 0.85–2.37; three studies; moderate certainty).

#### 3.4.4. Hispanic Ethnicity

The probability of waitlisting was not significantly affected by Hispanic ethnicity (OR: 1.09; 95% CI: 0.87–1.37; three studies; low certainty). Hispanic populations presented similar rates of overall kidney transplantation (OR: 1.10; 95% CI: 0.51–2.37; two studies; very low certainty), living-donor (OR: 1.22; 95% CI: 0.27–5.60; two studies; low certainty) and deceased-donor (OR: 1.23; 95% CI: 0.40–3.75; two studies; low certainty) transplantation compared to non-Hispanic White patients. Hispanic patients were estimated at a similar risk of graft failure (HR: 0.90; 95% CI: 0.68–1.20; five studies; low certainty) but were suggested to present significantly better survival rates (HR: 0.69; 95% CI: 0.66–0.71; two studies; high certainty).

#### 3.4.5. Female Sex

The waitlisting probability was significantly lower among female patients (OR: 0.89; 95% CI: 0.87–0.90; 14 studies; high certainty). Female sex was also associated with significantly decreased rates of overall kidney transplantation (OR: 0.91; 95% CI: 0.85–0.98; seven studies; moderate certainty), living-donor (OR: 0.88; 95% CI: 0.79–0.98; three studies; moderate certainty) and deceased-donor (OR: 0.94; 95% CI: 0.93–0.95; two studies; high certainty) transplantation. Preemptive transplantation was equally frequent between the two sexes (OR: 1.16; 95% CI: 0.79–1.71; two studies; low certainty). Female transplant recipients presented similar graft survival (HR: 0.99; 95% CI: 0.92–1.07; five studies; moderate certainty) but significantly better overall survival (HR: 0.87; 95% CI: 0.84–0.90; three studies; high certainty) compared to men.

#### 3.4.6. Low Socioeconomic Status

Patients with low socioeconomic status were estimated to have a lower probability of waitlisting for kidney transplantation (OR: 0.58; 95% CI: 0.51–0.66; 11 studies; moderate certainty) compared to patients with high socioeconomic status. The probability of receiving any kidney transplant was significantly decreased among those with low socioeconomic status (OR: 0.68; 95% CI: 0.58–0.79; eight studies; moderate certainty). Specifically, low socioeconomic status was linked to significantly lower rates of both living-donor (OR: 0.45; 95% CI: 0.38–0.54; four studies; moderate certainty) and deceased-donor transplantation (OR: 0.69; 95% CI: 0.54–0.89; three studies; low certainty). Graft failure risk was significantly higher among those with a low socioeconomic status (HR: 1.17; 95% CI: 1.07–1.28; 11 studies; moderate certainty); although, overall patient survival was similar between the compared groups (HR: 1.06; 95% CI: 0.91–1.23; 11 studies; low certainty).

### 3.5. Sensitivity Analysis

The analysis limited to studies conducted in the USA (Appendix A) indicated that Black patients presented significantly lower rates of successful AVF use, worse primary AVF patency and a lower probability of using peritoneal dialysis but better overall dialysis survival. In addition, Black Americans had a lower probability of being waitlisted and showed significantly lower rates of living-donor and preemptive transplantation, presenting worse graft survival. Compared to White patients, Asian patients with kidney failure were estimated to have lower rates of CVC use and home dialysis, better dialysis survival and a higher probability of being waitlisted. However, the rates of both living-donor and deceased-donor transplantation were significantly lower in Asian patients. Furthermore, dialysis survival was estimated to be better for Indigenous patients; although, they presented a significantly lower probability of being waitlisted and receiving a kidney allograft. Hispanic ethnicity was linked to better survival both among dialysis patients and kidney transplant recipients and was not associated with the probability of living-donor or deceased-donor transplantation. The female sex was associated with higher rates of CVC use and a lower likelihood of successful AVF use. Female patients were also estimated to have a lower probability of waitlisting and deceased-donor transplantation but significantly higher rates of preemptive transplantation. Dialysis survival was similar between the two sexes, while females presented better survival after kidney transplantation. Low socioeconomic status in the USA was associated with lower rates of transition from CVC to AVF or AVG, lower peritoneal dialysis use, worse dialysis survival, lower probability of waitlisting and kidney allograft receipt, as well as higher mortality after kidney transplantation.

## 4. Discussion

The present systematic review gathered the available literature knowledge and evaluated the association of race/ethnicity, sex and socioeconomic status with the management of kidney failure in adult patients. A total of 165 studies were included, exploiting data both from national registries and cohort studies. Meta-analyses were conducted, indicating remarkable disparities concerning patient survival, the choice of dialysis access and the probability of kidney transplantation.

The analysis proposed that racial and ethnic minorities present lower overall mortality rates, an observation that has been characterized as a “survival paradox”. In particular, Black and Asian patients, as well as patients of Hispanic ethnicity were linked to significantly better survival in dialysis compared to White patients, while Hispanic patients maintained the survival advantage even after kidney transplantation. This observation is in sharp contrast to the general population in which White individuals present a longer life expectancy. These discrepancies may partially reflect the existence of a survival bias, as Black patients ending up in dialysis tend to be healthier with favorable anthropometric characteristics and fewer comorbidities compared to White patients at the same stage [20]. Τhe healthy immigrant effect, as well as the potential under-reporting of deaths due to migration of the most vulnerable and elderly people to their place of origin, may also partly explain the observed better survival of those minorities [21]. Of note, Hispanic transplant recipients tend to be younger than White ones with lower rates of expanded-criteria donors, resulting thus in a better prognosis [22].

Black patients compared to White patients have also been associated with an increased risk of kidney failure, which some researchers have partly attributed to renal high-risk APOL1 variants [23]; hence, White patients ending up in kidney failure may have different comorbid conditions, associated with higher death risks, such as heart failure. As a result, a form of selection bias, namely collider bias, may complicate the interpretation of studies requiring the existence of a certain disease (i.e., kidney failure) as a prerequisite for patient inclusion [24]. Interestingly, Black patients with advanced kidney disease may be more likely to choose life-prolonging treatments, while White patients may prefer palliative care with more frequent withdrawal from dialysis [25]. It is important to note that Black patients show a more favorable survival than White ones in all age groups above 30 years, while the inverse relationship has been observed in patients 18–30 years old [26]. The increased mortality risk of younger Black patients may be due to the higher rates of fatal accidents, but also to the lower probability of health insurance and access to the health care system [27].

The successful use of AVF for dialysis was estimated to be less frequent among Black patients, a finding that may be explained to a certain extent by the delayed nephrology referral, as well as a generally smaller venous diameter, which is reflected by the lower probability of achieving a ≥4 mm diameter of the AVF postoperatively [28]. Black patients were also less likely to use peritoneal dialysis, possibly due to differences in physician and patient preferences [29], the lack of community support and potential language barriers precluding the proper education of patients about the benefits of this method [30].

Furthermore, Black patients had lower rates of waitlisting, preemptive and living-donor transplantation, which may be associated with various social factors, such as low socioeconomic status, poor psychosocial support and inadequate physician–patient communication [31]. From a medical point of view, African Americans have a greater probability of ABO incompatibility and a positive crossmatch test [32], while Black donors may be at a higher risk of developing chronic kidney disease, due to higher rates of hypertension and diabetes mellitus in conjunction with genetic factors [33]. Graft survival was estimated to be significantly worse among Black patients, which may be caused by a higher immunologic risk and a longer dialysis vintage, as well as by the lower bioavailability of calcineurin inhibitors due to P450-3A5 polymorphisms, predisposing to an increased rejection risk [34].

Indigenous patients presented similar choices of dialysis access, peritoneal dialysis use and overall survival compared to White patients. On the contrary, they showed significantly lower rates of waitlisting, living- and deceased-donor transplantation, both in North America and Oceania. Health inequalities for Indigenous people may arise from a variety of sociodemographic factors, including the combination of lower income and educational level, higher rates of behavioral risk factors for adverse health outcomes and the lack of access to culturally tailored health services [35]. Access to transplantation may be limited by social, language and cultural barriers that preclude the effective communication of Indigenous patients with healthcare professionals regarding their eligibility criteria, the benefits and risks associated with kidney transplantation [36]. As a result, Indigenous people tend not to complete the necessary steps for waitlisting and are often lost to follow-up [37]. It is important to note that Indigenous dialysis patients typically express a positive interest in receiving a kidney allograft since dialysis units are a considerable distance away from their place of residence and kidney transplantation offers them the opportunity to return to their homeland; however, Indigenous people remain practically uninformed about the transplantation steps and prospects, while they often are uncertain about their listing status. Residency in remote geographical areas with long distances from transplant centers also exacerbates the difficulty of accomplishing the pre-transplant evaluation [38]. Notably, Indigenous people living on large continents typically have to travel long distances to access healthcare services, while the existence of language diversity in colonized countries where the English language is embedded in the healthcare system limits their ability to fully understand professionals and educational material [39]. Several barriers may also limit kidney donation among Indigenous people, including the higher rates of comorbidities, in conjunction with cultural beliefs, health system mistrust and lower levels of health literacy leading to decreased rates of consent for donation [35].

The survival advantage of females in the general population is eliminated in dialysis (HR: 0.99; *p*-value: 0.824) but remains significant among kidney transplant recipients (HR: 0.87; *p*-value < 0.001). Among dialysis patients, female patients present higher mortality rates in the age group <45 years due to an increased infection risk [40], while in older ages, the cardiovascular risk of men rises gradually [41]. Female sex was associated with significantly higher rates of dialysis with CVCs, probably due to patient preferences associated with cosmetic reasons [42], as well as to the reluctance of nephrologists and surgeons about the maturation and long-term patency of AVFs in females. However, it should be noted that ultrasound studies have demonstrated that the female vasculature is equally adequate for the successful creation of an AVF [43]. Female patients presented also significantly lower rates of waitlisting and kidney transplantation, reflecting socioeconomic factors, potential physician biases regarding their suitability as transplant recipients [44], as well as the possible sensitization via the induction of alloantibodies in females with prior pregnancies [45].

Low socioeconomic status was associated with significantly higher mortality rates in dialysis; although, the certainty of the evidence was very low, mainly due to inconsistency in definitions and concerns of reporting bias. Furthermore, patients with low socioeconomic status presented worse primary AVF patency, less frequent peritoneal dialysis use, lower transplantation rates and increased graft failure risk. A low income may be linked to a lack of health insurance and limited access to medical care. These factors in conjunction with the possible residency in remote areas predispose to late nephrology referral, increasing the probability of starting dialysis with CVCs [46]. A low socioeconomic status may be accompanied by more comorbidities and unhealthy behaviors, such as smoking, that may prevent the successful maturation of an AVF. Moreover, a low education level may exacerbate the physician–patient communication barriers that need to be overcome in order to achieve an appropriate understanding of transplantation benefits and risks and promote optimal patient compliance [47].

The main strength of the present review relies on the inclusion of a large number of studies, assessing a plethora of dialysis- and transplantation-related outcomes. Large sample sizes were used in meta-analyses, thus achieving high precision. The GRADE approach was comprehensively applied, allowing a realistic evaluation of the quality of the available evidence in the field.

On the other hand, the interpretation of the results may be complicated by the inherent limitations of observational studies. Although the effect estimates of studies were derived from multivariate regression models, bias due to residual confounding cannot be excluded. Statistical heterogeneity was noticed in the majority of comparisons; hence, subgroup and sensitivity analyses were conducted to explore the potential sources of inconsistency. In particular, inter-study heterogeneity was evident in the evaluation of socioeconomic status; subgroup analysis was performed by categorizing studies depending on whether the socioeconomic status was determined by income, educational level or occupation, although significant variation in definitions was noted across studies. It is of note that the socioeconomic variables exert a differential impact on healthcare outcomes in different regions, complicating further the interpretability of the meta-analysis outcomes. In addition, the collection and use of data on racial and ethnic groups have inherent limitations since there is no national or international consensus for the definitions of race and ethnicity and societal definitions of racial and ethnic group membership, as well as individuals’ perceptions of their own racial and ethnic identities differing and changing over time. A person’s racial identification is therefore highly subjective, and consequently, each race is not actually a homogeneous group. This is particularly true for Hispanic people, who can be racially identified as White, Black, Asian, Native American or other ethnicities [48]. It should be also acknowledged that the meta-analysis was not designed to assess the potential interplay of the studied exposures, and thus, future research is needed to shed light on the influence of socioeconomic status on racial and ethnic differences in the health outcomes of kidney failure patients. Another important note is that evidence currently comes from high-income countries, while data from Asia and Africa remain sparse. Therefore, Asian patients were mainly evaluated in non-Asian regions, precluding the drawing of safe conclusions regarding dialysis and transplantation outcomes for Asian people worldwide. Differences in kidney failure outcomes may arise from sex-related biological differences, as well as from gender disparities linked to the social and cultural attributes of femininity [49]. However, it should be acknowledged that the potentially distinct associations of sex and gender with the investigated parameters were not separately evaluated in the majority of the included studies in this meta-analysis; hence, further research is needed to elucidate whether the observed differences between males and females reflect mainly sex-related differences or gender bias.

Racism and racial essentialism represent significant threats to health equity that should be recognized and mitigated by kidney professionals to promote social justice [50]. The findings of the study contribute to a deeper understanding of health inequities among kidney failure patients, forming the basis for the planning of treatment strategies to mitigate them. In this direction, organized programs for the management of patients with advanced kidney disease are needed, aiming to address issues concerning the choice of renal replacement therapy method, vascular access and access to kidney transplantation. Following such an approach, the treatment plan should be individualized to the patients’ preferences, functional status, current and future goals, life expectancy and degree of psychosocial support [51]. Moreover, organized educational programs using standardized material may be of value to target the mistrust of racial minorities, especially regarding the benefits of living-donor transplantation. Health policy interventions aiming to limit the impact of socioeconomic factors, such as low income and lack of insurance, would contribute to overcoming the barriers related to patient access to dialysis and transplantation, as well as ameliorating the quality of life and overall survival. Finally, the publication process of kidney research should ensure the equal representation of First Nations health research, aiming to limit systematic discrimination, promote justice and allow the development of culturally specific guidelines, tailored to the specific needs and perspectives of Indigenous people [52].

## 5. Conclusions

This meta-analysis indicated a survival paradox, with kidney failure patients of racial minorities presenting better survival rates. Black and female patients demonstrate decreased rates of successful dialysis via an arteriovenous fistula, as well as a lower probability of receiving a renal allograft. A low socioeconomic status predisposes to worse dialysis survival, more frequent use of central venous catheters and limited access to kidney transplantation. Future research is needed to elucidate the factors promoting the observed disparities and evaluate the health policy strategies that would allow equal access of patients to the optimal management and treatment options.

## Figures and Tables

**Figure 1 diseases-12-00023-f001:**
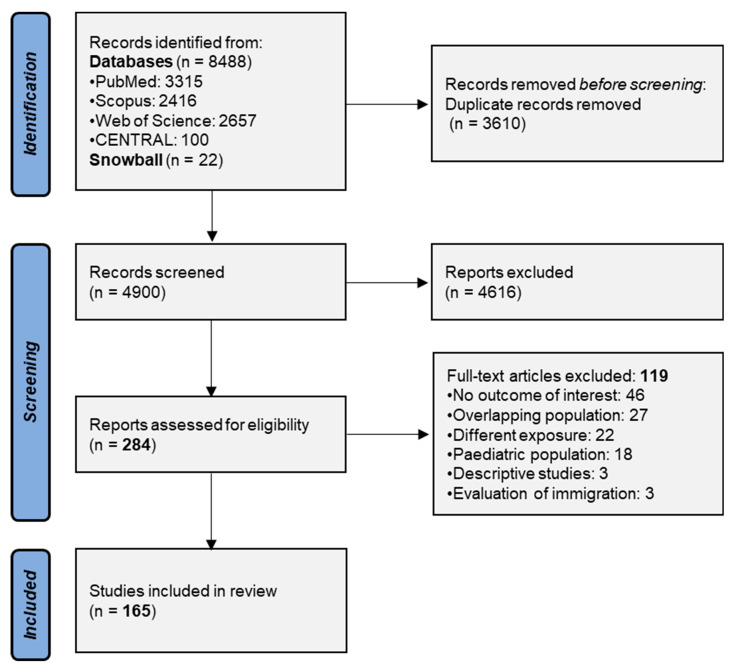
Search plot diagram.

**Figure 2 diseases-12-00023-f002:**
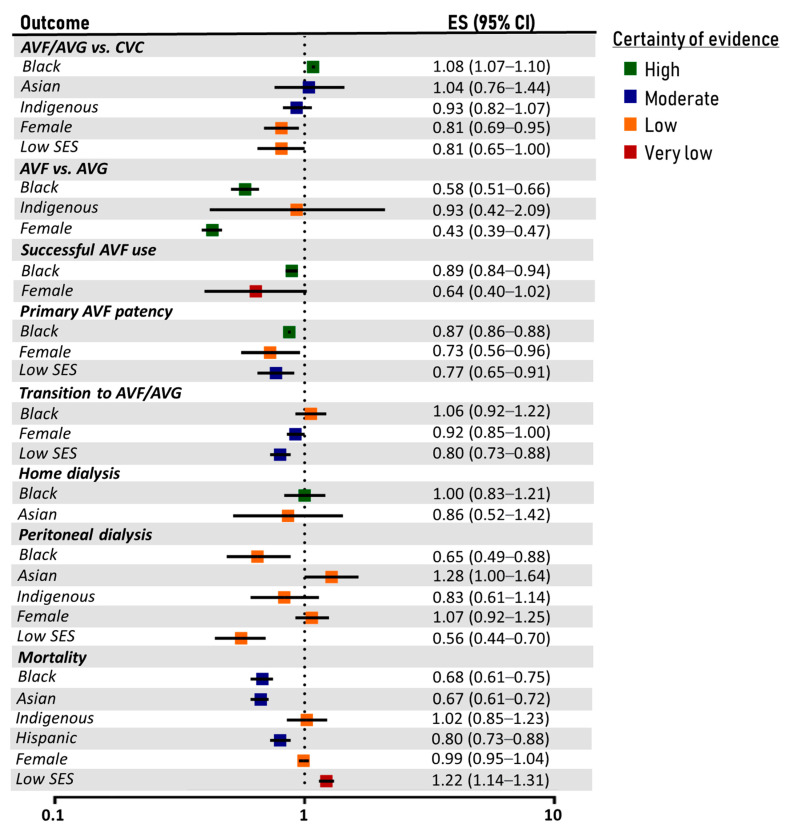
Forest plot and certainty of evidence of dialysis-related outcomes. White, male and high socioeconomic status patients served as the reference groups. Effect size corresponds to hazard ratio for primary AVF patency and mortality and to odds ratio for the other endpoints. ES: effect size; CI: confidence intervals; SES: socioeconomic status; AVF: arteriovenous fistula; AVG: arteriovenous graft; CVC: central venous catheter.

**Figure 3 diseases-12-00023-f003:**
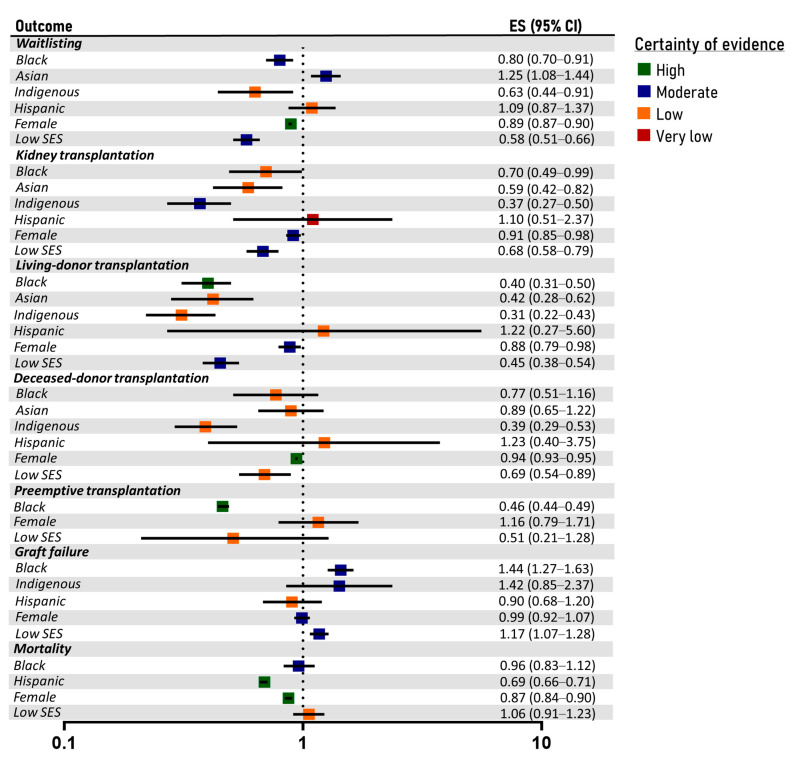
Forest plot and certainty of evidence of kidney transplantation-related outcomes. Effect size corresponds to hazard ratio for graft failure and mortality and to odds ratio for the other endpoints. White, male and high socioeconomic status patients served as the reference groups. ES: effect size; CI: confidence intervals; SES: socioeconomic status.

**Table 1 diseases-12-00023-t001:** Summary of study characteristics.

Characteristics	Included Studies
Overall	165
Country	
USA	70 (42.4%)
Canada	14 (8.5%)
United Kingdom	13 (7.9%)
Australia	11 (6.7%)
Design	
Prospective cohort	46 (27.9%)
Retrospective cohort	119 (72.1%)
Data source	
USRDS	31 (18.8%)
ANZDATA	12 (7.3%)
CORR	8 (4.8%)
Risk of bias	
Low	104 (63.0%)
Moderate	53 (32.1%)
Serious	8 (4.8%)
Population	
Hemodialysis	35 (21.2%)
Peritoneal dialysis	28 (17.0%)
Kidney transplant recipients	22 (13.3%)
Mixed	80 (48.5%)
Male sex	58.7 (54.6–61.3)% ^†^
Diabetes mellitus	38.0 (28.6–49.7)% ^†^

^†^ Median value (interquartile range). USRDS: United States Renal Data System; CORR: Canadian Organ Replacement Register. ANZDATA: Australian and New Zealand Dialysis and Transplant Registry.

## Data Availability

All extracted data are available as Appendix A.

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
