# Peer review of "Sociodemographic Disparities in Adults with Kidney Failure: A Meta-Analysis"

_diseases, 2024, doi:10.3390/diseases12010023_

Round 1

Reviewer 1 Report

Comments and Suggestions for Authors

Thank you for the opportunity to review this manuscript. This manuscript is not ready for publication. There are some serious issues that the authors need to address. The following are my comments describing these issues:

1.           The summary should indicate that it was registered in PROSPERO and indicate the registration code as indicated in the PRISMA 2020 guidelines.

2.           The Introduction is too long and does not focus on the topic studied nor does it completely justify the study.

3.           The second paragraph is very long and unfriendly for reading and understanding.

4.           Indicate with acronyms the reviewers and researchers who were in charge of study selection, data collection and quality assessment. Also, who was in charge of conflict resolution.

5.           Assess whether the GRADE methodology is applicable for a study that did not evaluate interventions.

6.           The figures should be improved. For example, Figure 3 shows in the heading ES and is defined as Effect size and this corresponds to an OR. Similarly, in the same figure HR is reported, which is confusing.

7.           In addition to the above, if the authors include observational studies, it would not be appropriate for them to use causal language using terms such as "effect".

8.           Clarify in Table 1 how they can determine that Male Sex corresponds to 58.7% this is an individual data or corresponds to the percentage of studies that included men.

9.           Clarify whether all studies calculated ORs or in some cases were RRs or are assuming RRs as ORs.

10.         The limitations of the study could be reviewed in order to assess whether there are any additional limitations. For example, the characteristics of the population could represent some characteristic that modifies the results or particular subgroup in relation to socioeconomic levels.

11.         Review that the PRISMA guideline criteria are actually met. There are several items that are not met. With this, an in-depth evaluation could be carried out, since all the minimum elements for reporting in a systematic review would be available.

12.         In line with the above, for example, not all search strategies and their results are included in the supplementary material.

Comments on the Quality of English Language

Minor editing of English language required.

Author Response

Reviewer #1:

Thank you for the opportunity to review this manuscript. This manuscript is not ready for publication. There are some serious issues that the authors need to address.

Response: Thank you very much for your review.

The following are my comments describing these issues:

  1. The summary should indicate that it was registered in PROSPERO and indicate the registration code as indicated in the PRISMA 2020 guidelines.

Response: Thank you for your comment. The PROSPERO registration code has been added at the end of the Abstract.

  1. The Introduction is too long and does not focus on the topic studied nor does it completely justify the study.

Response: Thank you for your comment. The Introduction section has been shortened and revised substantially focusing on the justification of the study.

  1. The second paragraph is very long and unfriendly for reading and understanding.

Response: The mentioned Introduction paragraph has been thoroughly revised and shortened. Specifically, the following concise text has been added: “Specifically, the risk of reaching kidney failure has been estimated to be significantly higher among Black and Hispanic patients by 3.4 and 1.3 times, respectively. In addition, Asian-Americans and Pacific-Islanders have been shown to present higher risk of early-stage kidney disease, while Native Hawaiian individuals have been proposed to be at greater risk of severe kidney disease forms. Racial discrimination may be associated with kidney disease progression through its potential effects on social determinants of health, such as access to education and healthcare, while it may exert biological impact by contributing to increased allostatic load and epigenetic changes that may be implicated in altered hormone metabolism, albuminuria and kidney function decline.”

  1. Indicate with acronyms the reviewers and researchers who were in charge of study selection, data collection and quality assessment. Also, who was in charge of conflict resolution.

Response: The acronyms of the reviewers have been added.

  1. Assess whether the GRADE methodology is applicable for a study that did not evaluate interventions.

Response: Thank you for the comment. The GRADE approach is applicable and has been widely used in meta-analyses of observational studies evaluating exposures (e.g., doi: 10.1016/j.eclinm.2022.101677, 10.1038/s41598-022-15443-9, 10.1016/j.jclinepi.2023.05.021, 10.1016/j.jclinepi.2023.03.003).

  1. The figures should be improved. For example, Figure 3 shows in the heading ES and is defined as Effect size and this corresponds to an OR. Similarly, in the same figure HR is reported, which is confusing.

Response: Thank you for your comment. The Figure legends have been improved, clarifying the endpoints in which the effect size corresponds to odds ratio or hazard ratio.

  1. In addition to the above, if the authors include observational studies, it would not be appropriate for them to use causal language using terms such as "effect".

Response: Thank you for your comment. The term “effect” has been replaced by terms of “association” throughout the text.

  1. Clarify in Table 1 how they can determine that Male Sex corresponds to 58.7% this is an individual data or corresponds to the percentage of studies that included men.

Response: Thank you for your comment. It has been clarified that it corresponds to the median percentage of male sex across studies, along with the interquartile range.

  1. Clarify whether all studies calculated ORs or in some cases were RRs or are assuming RRs as ORs.

Response: Thank you for your comment. In the Methods section, it has been specified that: “Odds ratios (OR) or hazard ratios (HR) along with their 95% confidence intervals (CI) were extracted from the multivariate models (where available) of original studies; in case multivariate models were not available or estimates were not directly reported, the crude OR or HR were extracted or calculated.”

  1. The limitations of the study could be reviewed in order to assess whether there are any additional limitations. For example, the characteristics of the population could represent some characteristic that modifies the results or particular subgroup in relation to socioeconomic levels.

Response: The limitations section has been enriched by adding the following statement: “It should be also acknowledged that the meta-analysis was not designed to assess the potential interplay of the studied exposures and thus future research is needed to shed light on the influence of socioeconomic status on racial and ethnic differences in the health outcomes of kidney failure patients.”

  1. Review that the PRISMA guideline criteria are actually met. There are several items that are not met. With this, an in-depth evaluation could be carried out, since all the minimum elements for reporting in a systematic review would be available.

Response: Thank you for your comment. A PRISMA checklist has been added as supplementary material (Appendix 10), specifying the elements that were reported in the systematic review.

  1. In line with the above, for example, not all search strategies and their results are included in the supplementary material.

Response: Thank you for your comment. In Appendix 1, the search algorithms along with the records obtained by each database are now reported.

Reviewer 2 Report

Comments and Suggestions for Authors

Thank you for the opportunity to review this article. The work is in line with the goals of the journal and effectively conveys the intended message. However, there are some elements that require additional attention from the authors. 

Abstract: Indices of low SES were associated with worse dialysis survival (HR: 1.22, 95% CI: 1.14-1.31)… I believe the HR value refers to mortality, not survival. I would suggest revising the sentence considering the value and meaning of the risk measure

In intro: The section on health disparities should be further revised to clarify their role in the outcomes and results of kidney failure. 

At line 47, the sentence about how racism impacts kidney pathophysiology is not clear and should be explicitly elaborated. It is perhaps better to explain that racism may worsen the condition (possibly by delaying diagnosis, hindering or delaying treatment, etc.) because it affects healthcare. However, this is not an impact on pathophysiology (i.e., disordered physiological processes associated with the disease). The authors should better clarify what they mean. 

What about the effect on other ethnicities? Again, they could provide better clarification on whether and what the effect is for, for example, Asians, etc.

In Figure 1, it is unclear how the authors report 4616 'reports not retrieved.' It is highly likely that they have misinterpreted the construction of the PRISMA flow diagram, specifically between 'not retrieved' and those excluded based on the initial t&a screening

Have the authors considered conducting a meta-regression to compare research findings and examine possible sources of heterogeneity across multiple studies?

Regarding the survival paradox, the relationship with different life expectancy should also be discussed

Comments on the Quality of English Language

A revision of spelling and sentence construction would be necessary

Author Response

Reviewer #2:

Thank you for the opportunity to review this article. The work is in line with the goals of the journal and effectively conveys the intended message. However, there are some elements that require additional attention from the authors. 

 Response: Thank you very much for your review.

Abstract: Indices of low SES were associated with worse dialysis survival (HR: 1.22, 95% CI: 1.14-1.31)… I believe the HR value refers to mortality, not survival. I would suggest revising the sentence considering the value and meaning of the risk measure.

Response: Thank you for your suggestion. The sentence has been revised as follows: “Indices of low SES were associated with higher mortality risk (HR: 1.22, 95% CI: 1.14-1.31)”.

In intro: The section on health disparities should be further revised to clarify their role in the outcomes and results of kidney failure. 

Response: Thank you for your comment. The Introduction section has been revised thoroughly as suggested, by adding details regarding the effects of race/ethnicity and racial discrimination on renal disease and kidney failure risk.

At line 47, the sentence about how racism impacts kidney pathophysiology is not clear and should be explicitly elaborated. It is perhaps better to explain that racism may worsen the condition (possibly by delaying diagnosis, hindering or delaying treatment, etc.) because it affects healthcare. However, this is not an impact on pathophysiology (i.e., disordered physiological processes associated with the disease). The authors should better clarify what they mean. 

Response: Thank you for your comment. The statement about the effects of racial discrimination has been revised as follows: “Racial discrimination may be associated with kidney disease progression through its potential effects on social determinants of health, such as access to education and healthcare, while it may exert biological impact by contributing to increased allostatic load and epigenetic changes that may be implicated in altered hormone metabolism, albuminuria and kidney function decline[7,8].”

What about the effect on other ethnicities? Again, they could provide better clarification on whether and what the effect is for, for example, Asians, etc.

Response: Thank you for your comment. The following text has been added: “Specifically, the risk of reaching kidney failure has been estimated to be significantly higher among Black and Hispanic patients by 3.4 and 1.3 times, respectively[5]. In addition, Asian-Americans and Pacific-Islanders have been shown to present higher risk of early-stage kidney disease, while Native Hawaiian individuals have been proposed to be at greater risk of severe kidney disease forms[6].”

In Figure 1, it is unclear how the authors report 4616 'reports not retrieved.' It is highly likely that they have misinterpreted the construction of the PRISMA flow diagram, specifically between 'not retrieved' and those excluded based on the initial t&a screening

Response: Thank you for your comment. Figure 1 has been revised accordingly.

Have the authors considered conducting a meta-regression to compare research findings and examine possible sources of heterogeneity across multiple studies?

Response: We thank the Reviewer for the comment. We have already conducted a meta-regression analysis that is mentioned in the Statistical analysis section as follows: Subgroup analysis (Appendix 7) was conducted based on study location, risk of bias, type of dialysis population and the definition of socioeconomic status. It has been further specified that: “P-values for subgroup differences were generated by conducting meta-regression analysis.” The outcomes of the subgroup analyses are already exhibited in the Results section.

Regarding the survival paradox, the relationship with different life expectancy should also be discussed.

Response: Thank you for your comment. It has been added in the text that the observed survival paradox is in sharp contrast to the general population in which White individuals present longer life expectancy.

Round 2

Reviewer 1 Report

Comments and Suggestions for Authors

None.

Comments on the Quality of English Language

Moderate editing of English language required.